# Spatial Methods for Inferring Extremes in Dengue Outbreak Risk in Singapore

**DOI:** 10.3390/v14112450

**Published:** 2022-11-04

**Authors:** Stacy Soh, Soon Hoe Ho, Annabel Seah, Janet Ong, Daniel R. Richards, Leon Yan-Feng Gaw, Borame Sue Dickens, Ken Wei Tan, Joel Ruihan Koo, Alex R. Cook, Jue Tao Lim

**Affiliations:** 1Environmental Health Institute, National Environment Agency, Singapore 138667, Singapore; 2Manaaki Whenua—Landcare Research, Lincoln P.O. Box 69040, New Zealand; 3Department of Architecture, College of Design and Engineering, National University of Singapore, Singapore 117566, Singapore; 4Saw Swee Hock School of Public Health, National University of Singapore and National University Health System, Singapore 117549, Singapore; 5Lee Kong Chian School of Medicine, Nanyang Technological University Novena Campus, Singapore 639798, Singapore

**Keywords:** dengue, extreme value theory, max-stable model, transmission risk

## Abstract

Dengue is a major vector-borne disease worldwide. Here, we examined the spatial distribution of extreme weekly dengue outbreak risk in Singapore from 2007 to 2020. We divided Singapore into equal-sized hexagons with a circumradius of 165 m and obtained the weekly number of dengue cases and the surface characteristics of each hexagon. We accounted for spatial heterogeneity using max-stable processes. The 5-, 10-, 20-, and 30-year return levels, or the weekly dengue case counts expected to be exceeded once every 5, 10, 20, and 30 years, respectively, were determined for each hexagon conditional on their surface characteristics remaining constant over time. The return levels were higher in the country’s east, with the maximum weekly dengue cases per hexagon expected to exceed 51 at least once in 30 years in many areas. The surface characteristics with the largest impact on outbreak risk were the age of public apartments and the percentage of impervious surfaces, where a 3-year and 10% increase in each characteristic resulted in a 3.8% and 3.3% increase in risk, respectively. Vector control efforts should be prioritized in older residential estates and places with large contiguous masses of built-up environments. Our findings indicate the likely scale of outbreaks in the long term.

## 1. Introduction

Infectious diseases commonly place the highest strain on health systems at the peak of outbreaks [1,2]. A large rise in the number of cases leads to an increased demand for healthcare services and potentially an exceedance of the capacity of healthcare institutions. Therefore, the appropriate management of infectious disease outbreaks is crucial for public health and outbreak preparedness. To achieve this, it is important to pre-empt outbreaks by forecasting and understanding their likely scale in order to mitigate their impending impacts [3].

Infectious disease control is conventionally augmented with methods of forecasting future epidemics. The ability to forecast accurately provides an early warning of an imminent outbreak and allows authorities to plan ahead and optimize resource allocation. Specific examples include the use of regression tools [4,5,6] or epidemic models [7,8] to predict future disease case counts or incidence. These models are suitable for anticipating and forestalling disease outbreaks but are unable to characterize the likely long-term behavior and scale of these outbreaks due to the forecasts being reliant on modeled conditional means and the occurrence of extreme events, such as the peak of disease outbreaks that are often fat-tailed in nature [9].

Modeling extreme events and the tail risk relies on the extreme value theory (EVT) framework, which makes use of appropriate statistics of extremes [9]. While several studies have used the EVT framework to model extremes in infectious disease transmission [1,10,11], few have considered exploiting spatial information to model extremes, with applications mainly limited to spatial extremes in weather phenomena such as precipitation [12], temperature [13], and wind speed [14]. In the context of infectious diseases, modelling their extremes across space can help policy makers formulate targeted interventions in areas with exceptionally high risk [1,2]. This is especially important for long-term resource planning and control efforts during outbreaks when resources for interventions are limited [3]. In this paper, we used Singapore as a case study to examine the utility of modeling the spatial risk of extreme dengue outbreaks in a densely populated city.

Dengue is a major vector-borne arboviral disease affecting much of the world, with an estimated 105 million infections annually [15]. Like most other countries in Southeast Asia, dengue is hyperendemic in Singapore, with all four serotypes in circulation [16,17]. This is attributed to the tropical climate, high connectivity with the rest of the world, and fully urbanized population, all of which facilitate the growth of dengue virus’s main vector, the *Aedes aegypti* mosquito [16,18]. Intense vector control efforts since the 1960s have sharply reduced the *Aedes* population in Singapore, which has led to a drop in the incidence of dengue. However, since the 1980s, low herd immunity has contributed to a resurgence of dengue in the city-state [19]. Like many other infectious diseases, dengue cases are often clustered spatially [20,21,22], as has been demonstrated in Singapore [23,24]. Differences in urban density, the age profile of buildings, and the presence or absence of natural vegetation contribute to spatial heterogeneity in *Ae. aegypti* distribution and, consequently, the spatial heterogeneity of dengue incidence [16].

Despite the importance of understanding disease transmission spatially, historical interest has largely been focused on the extreme behavior of the process without a consideration of spatial variation [2,25]. Standard EVT methods assume that observations are independent and identically distributed with no spatial autocorrelation. However, these assumptions render standard EVT methods inappropriate for characterizing extremes in the transmission of infectious diseases, which are often influenced by adjacent locations due to contagion. As such, we considered a class of extensions for standard EVT tools—max-stable processes—to characterize spatial heterogeneity in dengue extremes and allow for the characterization of its spatial dependency.

Here, we examined the utility of max-stable processes in analyzing the spatial distribution of extreme dengue outbreak risk in Singapore from 2007 to 2020. We accounted for possible associations between surface characteristics and dengue outbreak risk by modeling trend surfaces using said characteristics in the models. We then determined the return levels at different sites, which represented the weekly dengue case counts that were expected to be exceeded once every specific number of years, while taking into account changes in the trend surfaces.

## 2. Materials and Methods

### 2.1. Data

Dengue is a legally notifiable disease under the Infectious Diseases Act in Singapore, and the notification of all laboratory-confirmed cases to the Ministry of Health is mandated. We obtained data on all dengue infections aggregated by onset date from 2007 to 2020 from the Ministry of Health. Epidemiological information such as the residential addresses and onset dates of all laboratory-confirmed dengue cases were anonymized. We divided the land area of Singapore into regular hexagons, each with a circumradius of 165 m and an average area of 0.072 km^2^, and obtained the weekly counts of dengue infections for each hexagon. This radius was selected as it provided a fine enough resolution for data to be collected in Singapore, similar to [26]. We then obtained the annual maximum weekly counts of dengue cases for each hexagon between 2007 and 2020 inclusive (i.e., each block represented a year, and the block maximum was the maximum weekly dengue count in a particular year). We retained all hexagons with at least one instance of non-zero case counts during the study period. Hexagons with zero case counts throughout the study period were excluded from analysis, as these were non-residential areas and dengue transmission did not occur there.

Using land surface data mainly from 2018, we calculated the percentage of each land surface type in each hexagon. The land surface data was obtained from high-resolution satellite imagery from the Worldview 2 and 3 and Quickbird satellites [27]. These satellite data were classified into several land surface types, including: (i) marine, (ii) freshwater, (iii) impervious surfaces, (iv) non-vegetated pervious surfaces, (v) freshwater swamp and marsh, (vi) mangrove, (vii) vegetation with structure dominated by human management (with tree canopy), (viii) vegetation with structure dominated by human management (without tree canopy), (ix) vegetation with limited human management (with tree canopy), and (x) vegetation with limited human management (without tree canopy). We assumed that the land cover types remained constant throughout the study period.

Apartment blocks built by the Housing and Development Board (known as “public apartments”) make up the majority of residential housing in Singapore. We computed the median age of public apartments as of 2020 in each hexagon using the lease commencement year. We also obtained the aggregated population size in each hexagon from the Urban Redevelopment Authority (URA), Singapore. 

We obtained nationally representative climate data between 2007 and 2020 from Meteorological Services Singapore (MSS). We aggregated the daily climate readings collected by each of the 11 weather monitoring stations located across mainland Singapore by week, and computed the arithmetic mean values across all stations to derive overall weekly measures of mean, maximum, and minimum ambient temperature; mean relative humidity; and total rainfall. We derived weekly measures of absolute humidity (AH) from the mean temperature and relative humidity (RH) (see Appendix A). Weather data were included as temporal covariates in the max-stable models (Appendix A).

### 2.2. Max-Stable Processes

Consider observations of infectious disease case counts $I_{1:t,j}=\left\{I_{1,j},\;I_{2,j},\;\ldots ,\;I_{t-1,\;j},\;I_{t,j}\right\}$ measured at some fixed interval for time points 1 to *t*, and locations *j* = {1, 2, …, *J*}. Our goal was to model the peak of dengue outbreak risk from tools derived from EVT, which allowed us to group our observations into separate blocks denoted by *n*_1_, *n*_2_, …, *n_m_* for some fixed size *N* to yield the block maxima set of observations $z_{1:n_{m}}$ by taking the largest order statistic, denoted by the subscript (*N*) in each block:$$I_{1:t,\;j}=\left\{I_{1,\;j},\;I_{2,\;j},\;\ldots ,\;I_{t-1,\;j},\;I_{t,\;j}\right\}\;$$
$$\;=\left\{I_{1,\;j,\;n_{1}},\;I_{2,\;j,\;n_{2}},\;\ldots ,\;I_{k-1,\;j,\;n_{1}},\;I_{k,\;j,\;n_{2}},\;I_{k+1,\;j,\;n_{2}},\;\ldots ,\;I_{t-2,\;j,\;n_{m}},\;I_{t-1,\;j,\;n_{m}},\;I_{t,\;j,\;n_{m}}\right\}\;$$
$$\;=\left\{I_{n_{1},\;j},I_{n_{2},\;j},\;\ldots ,\;\;I_{n_{m},\;j}\;\right\}\;$$
$$z_{1:n_{m,\;j}}=\{I_{n_{1},\;\left(N\right),\;j\;},\;I_{n_{2},\;\left(N\right),\;j\;},\ldots ,I_{n_{m},\;\left(N\right),\;j\;}\}\;$$
where $z_{1:n_{m,\;j}}$ represents the block maxima set of observations.

Max-stable processes are widely used for modeling spatial extremes, such as the observations $z_{1:n_{m,\;j}}\;$ above. Briefly, let ${\left\{S_{j}\right\}}_{j=1}^{\infty }$ be the points of a Poisson process on ℝ+ with intensity *ds/s*^2^, and let ${\left\{W_{j}\left(z\right)\right\}}_{j=1}^{\infty }$ be independent replicates of some stationary process *W*(*x*) to be ascertained later, satisfying the condition $E\left[\max\left\{0,W_{j}\left(o\right)\right\}\right]=1$, where *o* denotes the origin. We can then define:(1)$$Z\left(x\right)=\underset{j}{\max}\;S_{j}max\left(0,\;W_{j}\left(x\right)\right)\;$$
where *Z*(*x*) is a max-stable process and $z=z_{1:n_{m,\;j}}$ represents the block maxima set of observations. In our study, $S_{j}$ was interpreted as governing the probability of observing the maximum weekly dengue case count at a particular location *j*, while $W_{j}\left(x\right)$ governed the spatial relationship of all the (maximum) weekly dengue case. Different choices for the process $W\left(x\right)$ resulted in different max-stable models. We explored three commonly used max-stable models, namely the Smith, Schlather, and Brown–Resnick models, for modeling spatial extremes of dengue. The restrictions they impose on $W_{j}\left(x\right)$ are presented in Table 1 and outlined in the following paragraphs.

#### 2.2.1. Max-Stable Models: Smith Model

Firstly, we took $W_{j}\left(x\right)=g\left(x-X_{j}\right)$, where *g* denotes a probability density function to be defined and $X_{j}$ a homogeneous Poisson process. We could interpret the value of the max-stable process at *x* as the maximum over a space of the maxima of dengue outbreaks, centered at the random points $X_{j}$ and of magnitude $S_{j}$. The effects at *x* were given by $S_{j}g\left(x-X_{j}\right)$. We arrived at the Smith model when taking g to be a (multivariate) normal distribution, with covariance matrix *Σ* [28,29].

Then, considering any two magnitudes $z_{1}$ and $z_{2}$ for dengue outbreaks in locations $x_{1}$ and $x_{2}$, respectively, we obtained the following bivariate cumulative distribution function parameterizing their spatial dependence structure under the Smith model:(2)$$\mathrm{Pr}\left\{Z\left(x_{1}\right)\leq z_{1},\;Z\left(x_{2}\right)\leq \;z_{2}\right\}=\exp[-\frac{1}{z_{1}}\Phi \;(\frac{a}{2}+\frac{1}{a}\log\frac{z_{2}}{z_{1}})-\frac{1}{z_{2}}\;\Phi \;(\frac{a}{2}+\frac{1}{a}\log\frac{z_{1}}{z_{2}})]\;\;$$
where $a^{2}={\left(x_{1}-x_{2}\right)}^{\prime }\;\sum (x_{1}-x_{2})$ is the Mahalanobis distance between locations $x_{1}$ and $x_{2}$, and Φ denotes the standard normal cumulative distribution function.

#### 2.2.2. Max-Stable Models: Schlather Model

The Schlather model considers a more flexible class of max-stable processes by taking $W_{j}\left(x\right)=\sqrt{2\pi }\;max\left(0,\;\varepsilon_{j}\left(x\right)\right)$, where $\varepsilon_{j}$ are independent copies of a standard Gaussian process with correlation function ρ. The bivariate cumulative distribution function is given by: (3)$$\mathrm{Pr}\left\{Z\left(x_{1}\right)\leq z_{1},\;Z\left(x_{2}\right)\leq z_{2}\right\}=\exp[-\frac{1}{2}\;\left(\frac{1}{z_{1}}+\frac{1}{z_{2}}\right)(1+\sqrt{1-\frac{1+\rho \left(x_{1}-x_{2}\right)\;z_{1}\;z_{2}}{{\left(z_{1}+z_{2}\right)}^{2}}}\;)]\;$$
where flexibility may be obtained from the various correlation functions ρ allowed under the Gaussian process formulation.

#### 2.2.3. Max-Stable Models: Brown–Resnick Model

Lastly, the Brown–Resnick model takes $\;W_{j}\left(x\right)=exp\left\{\varepsilon_{j}\left(x\right)-\sigma^{2}\left(x\right)/2\right\}$, where $\varepsilon_{j}$ are independent copies of a centered Gaussian process with stationary increments, such that $var\left\{W\left(x\right)\right\}=\sigma^{2}\left(x\right)$ for all *x ∈ X*. Then, considering any two magnitudes $z_{1}$ and $z_{2}$ for dengue outbreaks in locations $x_{1}$ and $x_{2}$, respectively, their bivariate cumulative distribution is conveniently given by (2), but with $a^{2}=var\left\{W\left(x_{1}-x_{2}\right)\right\}$. Max-stable models were fitted using the pairwise likelihood approach, which maximized the weighted sum of likelihoods between all possible location pairs under our specifications, as described in [28,29].

#### 2.2.4. Modeling Trend Surfaces

First, note that each pointwise distribution of *Z* is the generalized extreme value distribution of the form:(4)$$\mathrm{Pr}\left(Z\left(x\right)\leq z|µ,\;\sigma^{*},\;\xi \right)=exp\left(-{\left(1+\xi \frac{z-µ\;}{\sigma^{*}}\right)}^{-1/\xi }\right),\;\left(1+\xi \frac{z-µ\;}{\sigma^{*}}\right)>0$$
where *µ*, *σ^∗^*, and *ξ*, respectively, denote the location, scale, and shape parameters [29]. We further incorporated the spatial dependence of dengue outbreaks through other covariates for each location by fitting each pointwise GEV distribution using their respective parameters (location *µ*, scale *σ^∗^*, and shape *ξ* parameters), as well as other related spatial characteristics, through the following linear equations [28,29]:$$µ\left(x\right)=\beta_{µ,\;0}+\beta_{µ,\;1}Q_{1}\left(x\right)+\cdots +\beta_{µ,\;q}Q_{q}\left(x\right)$$
$$\sigma^{*}\left(x\right)=\beta_{\sigma^{*},\;0}+\beta_{\sigma^{*},\;1}Q_{1}\left(x\right)+\cdots +\beta_{\sigma^{*},\;q}Q_{q}\left(x\right)\;$$
$$\xi \left(x\right)=\beta_{\xi ,0}$$
where $Q_{1:q}\left(x\right)$ represents the respective spatial observations which may influence z linearly through the regression coefficients $\beta_{1:q}$ at location *x*. We wrote $\beta_{\text{µ},\sigma *,\xi ,0}$ to denote the respective intercept terms for each marginal GEV parameter and restricted $\xi \left(x\right)$ to a spatial constant due to the large uncertainty surrounding the parameter as per the existing literature [29]. We considered a large group of spatial parameters which have been hypothesized to either affect dengue transmission dynamics or vector ecology [16]. The pointwise structure was chosen using pairwise deviance and empirical characteristics of local data after fitting a wide range of plausible models.

The *r*-year return level for location *x* denoted ${\hat{z}}_{r}\left(x\right)$ could then be estimated by inverting (4) and computing the following using the estimated parameters $\left\{\hat{µ}\left(x\right),\;{\hat{\sigma }}^{*}\left(x\right),\;\hat{\xi }\right\}$:(5)$${\hat{z}}_{r}\left(x\right)=\;\hat{µ}\left(x\right)+\;\frac{{\hat{\sigma }}^{*}\left(x\right)}{-\hat{\xi }}\left[{\left(-\log\left(1-r^{-1}\right)\right)}^{-\hat{\xi }}-1\right]$$
where ${\hat{z}}_{r}\left(x\right)$ provides the estimated weekly dengue case counts expected to be exceeded once every *r* years at location *x* and can be interpreted as the long-term extreme risk associated with dengue. To further determine the impact of some spatial covariate on the return level, we assumed that the covariate $Q_{h}$ increases by some *w* unit. Given the estimated parameters for the regression equations, we computed the new values for the GEV parameters at each location post hoc:$$µ{\left(x\right)}^{\prime }={\hat{\beta }}_{µ,0}+{\hat{\beta }}_{µ,1}Q_{1}\left(x\right)+\cdots +{\hat{\beta }}_{µ,j}(Q_{h}\left(x\right)+w)+\cdots +{\hat{\beta }}_{µ,q}Q_{q}\left(x\right)\;$$
$$\sigma {\left(x\right)}^{\prime }={\hat{\beta }}_{\sigma^{*},0}+{\hat{\beta }}_{\sigma^{*},1}Q_{1}\left(x\right)+\cdots +{\hat{\beta }}_{\sigma^{*},j}(Q_{h}\left(x\right)+w)+\cdots +{\hat{\beta }}_{\sigma^{*},q}Q_{q}\left(x\right)\;$$

The hypothetical *r*-year return level given the increase in the respective covariate could then be taken as:(6)$${\hat{z}}_{r}{\left(x\right)}^{\prime }=\;\hat{µ}{\left(x\right)}^{\prime }+\frac{\hat{\sigma }{\left(x\right)}^{\prime }}{-\hat{\xi }}\left[{\left(-\log\left(1-r^{-1}\right)\right)}^{-\hat{\xi }}-1\right]\;$$
and compared against estimates obtained using (5). For exposition, we examined the impact of a 1%, 5%, and 10% increase in the land cover proportions and the total population on the 30-year return level from the final model used. Similarly, for the median age of public apartments, we examined the impact of a 1, 2, and 3 year increase in age on the 30-year return level relative to the baseline.

#### 2.2.5. Model Assessment

We considered various permutations of marginal pointwise structures under different max-stable models using pairwise deviance as the model selection criteria. The max-stable fitting procedure substituted the full likelihood for the pairwise likelihood due to the large number of locations present in the spatial data [30]. The pairwise deviance was therefore computed under the maximum pairwise likelihood estimator. After fitting the max-stable process, we assessed if the marginal distributions were appropriately modeled and whether the spatial dependence structure was satisfactory in the final model. We conducted model assessment in two ways: with a quantile–quantile (QQ) plot (Appendix A) and an extremal coefficient estimates plot from the F-madogram (Appendix A). The QQ plots compared the observed pairwise maxima for each block and those obtained by simulations from the fitted model. To further assess if spatial dependence was adequately modeled in the final model, we empirically estimated the pairwise extremal coefficient between each pair of sites from the binned F-madogram. The extremal coefficient function θ(.) is a statistical tool used to diagnose the dependence in the context of max-stable processes, where θ(h) is defined as the measure of the dependence of a pair of sites separated by the distance h [28].

#### 2.2.6. Model Fitting

We fitted max-stable models using the R package ”SpatialExtremes” [31]. Three types of models, the Smith, Schlather, and Brown–Resnick models, were fitted, and a variety of specifications for each of their parameters was tested. The location *µ* and scale *σ* parameters of each model were allowed to vary spatially via the inclusion of trend surfaces and be influenced by the land cover covariates from the satellite images (Appendix A) [27]. The specifications of the four most favorable models are provided in Table 2.

## 3. Results

### 3.1. Descriptive Results

A total of 1618 hexagons were included for analysis; these contained at least one instance of non-zero dengue case counts from 2007 to 2020. The distribution of the 1618 block maxima by year is shown in Table 3. To visualize the spatial distribution of block maxima, we calculated the average block maxima in each hexagon between 2007 and 2020 before plotting those averages on a map. The average block maxima ranged from 0 to 9 cases (to the nearest whole number) with a mean of 1.3 cases (Figure 1A).

There were varying proportions of the different types of land cover throughout Singapore. Built-up areas (impervious surfaces) occupied most of the main island (Figure 1C), comprising an average proportion of 30% in every hexagon, but trees and other forms of vegetation were also common in those places (Figure 1E,F) and made up nearly 20% collectively. Forests (vegetation with a tree canopy under limited human management) were largely confined to the extreme west and center of the main island and on major outlying islands to the northeast (Figure 1G), making up at least half of each spatial unit in those areas specifically. The median age of public apartments (Figure 1I) was 14 years and was higher in the east. The population size (Figure 1J) averaged 1380 per spatial unit and was also higher in the east compared to other parts of Singapore. The spatial distribution of reported dengue cases largely followed that of residential areas, as expected given that *Ae. aegypti* is a peri-domestic species.

### 3.2. Model Assessment

The Brown–Resnick max-stable model best characterized the extreme dengue outbreak risk.

The four best max-stable models were selected, together with their corresponding marginal pointwise structures using the model selection criteria of pairwise deviance (Table 2). We obtained the model with the lowest pairwise deviance under a Brown–Resnick process (M1). The next two most favorable models were Schlather models (M2, M3) under a powered exponential correlation function with varying pointwise structures (relative deviance: M2, 1.007; M3, 1.022). The fourth most favorable model followed another Brown–Resnick process with a different pointwise structure (relative deviance: M4, 1.427). We also considered various permutations of marginal pointwise structures under different max-stable models (see Appendix A). In the following paragraphs, we refer to the model with the lowest pairwise deviance (M1) unless stated otherwise.

### 3.3. Estimated Extreme Dengue Outbreak Risk across Space and Associated Spatial Drivers

Under the four most favorable model specifications (M1–M4), we generated the return levels for return periods of 5, 10, 20, and 30 years for all hexagons included in the analysis (Figure 2). The return levels provided the level (i.e., number of weekly dengue cases) that was expected to be exceeded once in a pre-specified return period in a particular hexagon, which indicated the extreme dengue outbreak risk. Across all four models, the return levels were consistently highest in the eastern parts of Singapore, encompassing districts such as Ang Mo Kio, Hougang, Bedok, and Tampines, thereby indicating larger extreme dengue outbreak risk in these areas. Unsurprisingly, the return levels increased with the duration of the return period. 

Under M1, the ranges for the estimated return levels for the 5-, 10-, 20-, and 30-year return periods were 2–15, 4–32, 9–64, and 13–92, respectively, per hexagon (Figure 2). The estimated return levels over the 5-, 10-, and 20-year periods across all model specifications (M1–M4) were largely similar, with the upper limit ranges being 11–23, 24–37, and 49–64, respectively. The 30-year return levels diverged across model specifications to a larger extent, with the upper limit ranging between 65 and 96 (Figure 2).

Under M1, one would expect the maximum weekly dengue cases to exceed the mean return level of 51 cases at least once in 30 years in most of eastern Singapore. Conversely, the return levels associated with the western parts of Singapore were lower, ranging mostly between 13 and 45 cases per hexagon. Under M1, we obtained the spatially varying marginal parameters of the trend surfaces across hexagons, i.e., the location and scale parameter (see Appendix A). The shape parameter was assumed to be constant given the absence of a clear spatial pattern as well as the difficulty of estimating this parameter [28]. While no distinct pattern could be discerned for the location parameter, the spatial distribution of the scale parameter followed closely that of the return levels, with higher outbreak risk in the east.

Land cover factors such as the percentages of freshwater, impervious surfaces, and vegetation with human management and tree canopies in each hexagon, as well as the population density and the age of public apartments, were associated with higher return levels. For each hexagon, we computed the independent effect of a 1, 5, and 10% increase in individual land cover factors on the return level expected for a 30-year return period under M1 for the following factors: (i) percentage of freshwater, (ii) percentage of impervious surfaces, (iii) percentage of vegetation with structure dominated by human management with tree canopy, and (iv) population density in each hexagon. Additionally, we computed the effect of a 1-, 2-, and 3-year increase in the age of public apartments on the 30-year return levels in each hexagon. The independent effects of a unit change in these spatial factors on the 30-year return levels are presented in Table 4 and Figure 3. The age of public apartments and the percentage of impervious surfaces had the largest impact on the size of the return levels. The percentage of vegetation and freshwater surfaces and the population size also influenced the return levels but to a smaller extent. 

## 4. Discussion

An increase in extreme dengue outbreak risk is likely to have a disproportionately large impact on public health resources. Our work adds to the growing literature on dengue risk modeling and mapping in Singapore by quantifying the scale and likelihood of extreme dengue outbreak risk across the country. Given the localized nature of dengue outbreaks, spatial heterogeneity has to be appropriately accounted for [24]. We achieved this through the use of max-stable models and identified potential spatial factors that could drive the risk of extreme dengue outbreaks. The incorporation of trend surfaces further allowed for spatial variation in our model parameters. 

Our results demonstrate that the eastern regions of Singapore, roughly corresponding to the eastern third of the main island, have higher return levels compared to the west in general, indicating that these regions have higher extreme dengue outbreak risk. This could be explained by the higher population density and older age of public apartments in this area and is supported by the higher incidence of dengue observed in these regions. Our analysis indicated that the age of public apartments and the proportion of impervious surfaces had the strongest influence on return levels. Under M1, a 3-year increase in the median age of public apartments was associated with an average 3.8% increase in the 30-year return level, while a 10% increase in the percentage of impervious surfaces was associated with an average 3.3% increase in the 30-year return level. The areas with a higher risk of dengue outbreak were located in the east, with the return levels remaining highest in these regions. Older public apartments are likely to exhibit increased infrastructural degradation over time, which is conducive for water stagnation, providing more breeding habitats for *Aedes* mosquitoes [32]. The amount of impervious surfaces is a proxy for urban residential density [33], which provides favorable breeding conditions for *Ae. aegypti*. Impervious surfaces, which absorb greater amounts of heat than pervious ones during the day and release heat more slowly at night, are positively associated with the effective reproduction number of dengue in Singapore [34]. Additionally, these surfaces have been associated with vectorial capacity in Athens, Georgia, US [35]. These results are to be expected given that the main vector for dengue, *Ae. aegypti*, is a peri-domestic species that thrives in urban environments. It is therefore prudent for public health agencies to focus their vector control efforts on older residential estates and localities where there are large contiguous masses of built-up environments, as found in the eastern parts of Singapore.

We also considered how different land surface types could have influenced the extreme dengue outbreak risk. The percentages of vegetation structurally dominated by human management and tree canopies and freshwater surfaces, as well as the population size, were associated with higher return levels. Vegetation and shrubs provide natural habitats for the secondary dengue vector, *Ae. albopictus* [36], while freshwater surfaces are often located near or within nature reserves in Singapore, which provide natural outdoor habitats for *Ae. albopictus* and facilitate mosquito breeding. The size of the human population is also associated with higher return levels. The close proximity between humans and *Aedes* mosquitoes in areas with a higher population density encourages the mosquitoes to consume blood more frequently. Taken together, these are factors which contribute to favorable vector breeding conditions and the transmissibility of the dengue virus, thereby exacerbating the estimated extreme dengue outbreak risk across localities.

EVT is gaining currency in infectious diseases modeling. It has been applied to weekly rates of pneumonia and influenza [2], the historical distribution of pandemics [9], and superspreading events for respiratory infections [37]. The fat-tailed nature of disease outbreaks demonstrate that the use of statistical methods based on analyses of means to model such phenomena have their limits, and that it would be more appropriate to utilize statistics of extremes. Spatial heterogeneity is another characteristic of infections, with different disease dynamics occurring at different locations dependent on variations in the distribution of pathogens, vectors, and hosts [38,39]. Together with [1,40], our work is one of the first to apply EVT to vector-borne disease outbreak risk and, to the best of our knowledge, the first to incorporate spatial heterogeneity in modeling extreme dengue outbreak risk in Singapore. We note that there are other methods for measuring the spatial dependence of infectious disease cases. For instance, the spatiotemporal tau statistic is a non-parametric measure of the degree of clustering at a specific range of distances (d1, d2) from a particular case, specifically, the ratio of the expected incidence rate of cases falling within (d1, d2) to the average incidence rate across the entire population [41,42]. Another method of modeling the spatial dependence of infectious diseases is Bayesian inference through the integrated nested Laplace approach (INLA) [43]. Like max-stable models, the spatial dependence in INLA is explicitly built into the equations for the hyperparameters. While these and several other methods [41] are useful in modeling the spatial dependence of infectious diseases, they do not focus on the extremes of transmission risk, which are best analyzed using the framework of EVT and max-stable models. Our models were able to infer the return values, which represented the extreme dengue outbreak risk, at specific locations and varying durations into the future. Despite Singapore being a small country, we showed that spatial disparities in extreme dengue outbreaks were driven by variations in land surface types and other aspects of the built environment such as building age. This demonstration of city-scale variation in extreme dengue outbreak risk, and the possible factors associated with it, is relevant to other metropolitan areas in dengue-endemic regions. Our work also demonstrates the importance of spatially resolved dengue surveillance systems, as they may help public health agencies triangulate the implications of a changing city landscape on extreme dengue outbreak risk and help guide long-term resource and policy planning.

Nevertheless, there are several limitations to our study. Firstly, as a standalone measure, the return level might not provide a complete picture of dengue outbreak risk, since it focuses only on the block maxima of a time series, which in our case is the maximum weekly dengue cases in a year. The return levels could therefore be used in conjunction with other forecasting tools that examine the mean levels of disease transmission. Secondly, our analysis was limited to locations where dengue cases were recorded between 2007 and 2020. As a result, we did not achieve the full spatial coverage of Singapore, nor were we able to extrapolate the outbreak risk to those areas. However, this limitation is mitigated by the fact that no dengue transmission occurred in these areas during the period of study, since they are not residential areas. Thirdly, the surface land cover information was based on a single year, 2018, and was assumed not to vary over time. While this was a reasonable assumption within our study period of 2007 and 2020, given the land-use planning guidance in Singapore [44], it might not hold true several decades into the future if land uses change. Lastly, we did not consider human movement in our study, which might have influenced the transmission risk of dengue. Future studies may consider incorporating this aspect in the modeling of extreme dengue risk. 

## 5. Conclusions

To our knowledge, this is the first study to model the spatial risk of extreme dengue outbreaks in Singapore. The eastern parts of Singapore were found to be more prone to extreme dengue outbreak risk, which is consistent with the observed spatial clustering of dengue cases. These findings provide a more comprehensive understanding of how extreme outbreaks might be clustered across Singapore and the risk of such events. We also identified spatial factors that might drive extreme dengue outbreak risk. Our findings can be used to inform long-term resource planning and risk mitigation in Singapore.

## Figures and Tables

**Figure 1 viruses-14-02450-f001:**
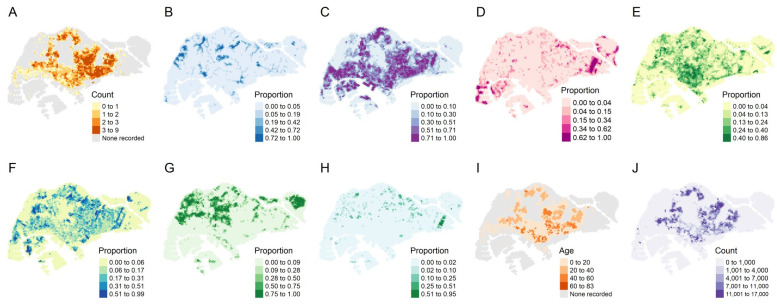
Top row from left to right: (**A**) average yearly maximum weekly dengue case counts from 2007 to 2020, (**B**) proportion of freshwater surfaces, (**C**) proportion of impervious surfaces, (**D**) proportion of non-vegetated pervious surfaces, and (**E**) proportion of vegetation with structure dominated by human management with tree canopy. Bottom row from left to right: (**F**) proportion of vegetation with structure dominated by human management without tree canopy, (**G**) proportion of vegetation with limited human management with tree canopy, (**H**) proportion of vegetation with limited human management with tree canopy, (**I**) median age of public apartments, and (**J**) population size in 2020.

**Figure 2 viruses-14-02450-f002:**
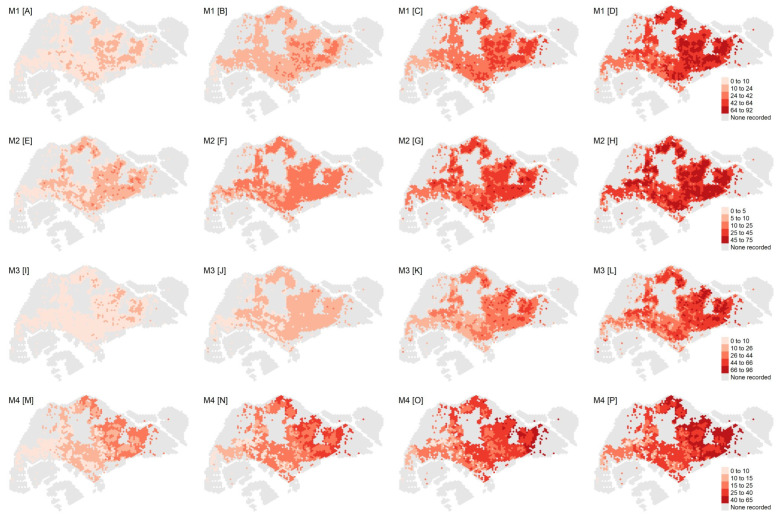
Top row from left to right: return levels for M1 associated with a (**A**) 5-year, (**B**) 10-year, (**C**) 20-year, and (**D**) 30-year return period. Second row from left to right: return levels for M2 associated with a (**E**) 5-year, (**F**) 10-year, (**G**) 20-year, and (**H**) 30-year return period. Third row from left to right: return levels for M3 associated with a (**I**) 5-year, (**J**) 10-year, (**K**) 20-year, and (**L**) 30-year return period. Fourth row from left to right: return levels for M4 associated with a (**M**) 5-year, (**N**) 10-year, (**O**) 20-year, and (**P**) 30-year return period. The legend for the return level of each row is shown on the right.

**Figure 3 viruses-14-02450-f003:**
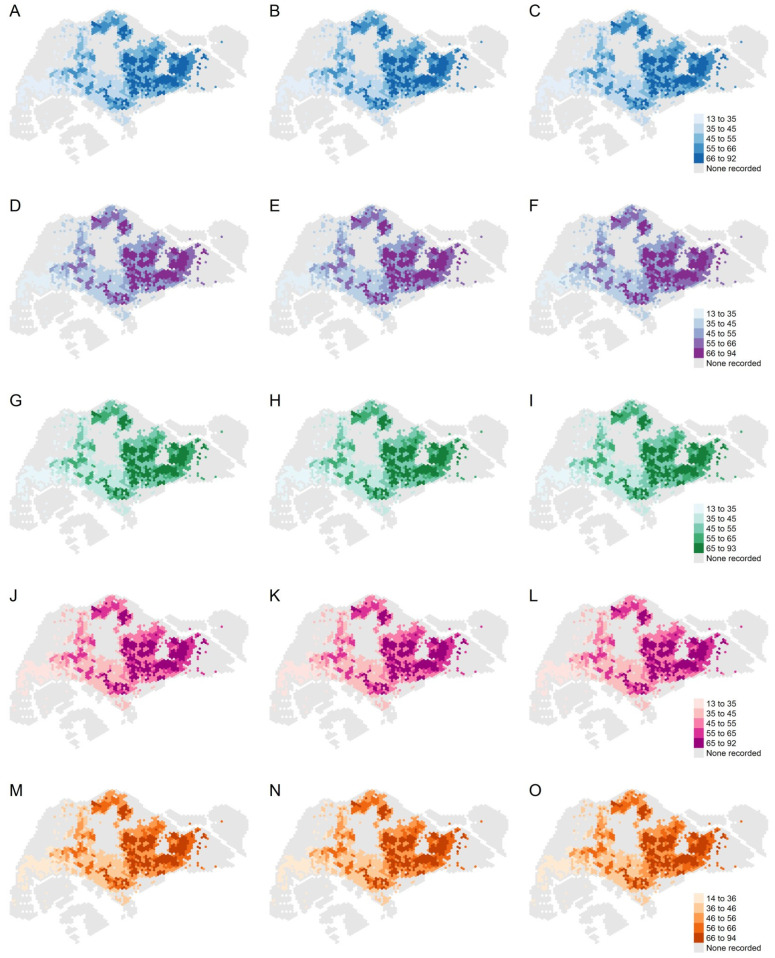
Top row from left to right: 30-year return levels under M1 with a (**A**) 1 percent, (**B**) 5 percent, and (**C**) 10 percent increase in freshwater surfaces. Second row from left to right: 30-year return levels under M1 with a (**D**) 1 percent, (**E**) 5 percent, and (**F**) 10 percent increase in impervious surfaces. Third row from left to right: 30-year return levels under M1 with a (**G**) 1 percent, (**H**) 5 percent, and (**I**) 10 percent increase in vegetation with human management with tree canopy. Fourth row from left to right: 30-year return levels under M1 with a (**J**) 1 percent, (**K**) 5 percent, and (**L**) 10 percent increase in the population. Fifth row from left to right: 30-year return levels under M1 with a (**M**) 1-year, (**N**) 2-year, and (**O**) 3-year increase in the median age of public apartments. The legend for the return level of each row is shown on the right.

**Table 1 viruses-14-02450-t001:** Max-stable models and corresponding restrictions on stochastic function. ***GP*** denotes a Gaussian process.

Model	Restriction
Smith	$W_{j}\left(x\right)=g\left(x-X_{j}\right)$
Schlather	$W_{j}\left(x\right)=\;\sqrt{2\pi }\;max\left(0,\;\varepsilon_{j}\left(x\right)\right),$ where $\varepsilon_{j}\left(x\right)\;\sim \;GP\left(0,\rho \right)$
Brown–Resnick	$W_{j}\left(x\right)=exp\left(\varepsilon_{j}\left(x\right)-\sigma^{2}\left(x\right)/2\right),\;$ where $\varepsilon_{j}\left(x\right)\;\sim \;GP\left(0,\rho \right)$

**Table 2 viruses-14-02450-t002:** Pairwise deviance from the four most favorable max-stable models.

Model ^1^	Trend Surfaces ^2^	Relative Deviance ^3^
Brown–Resnick **(M1)**	$µ\left(x\right)=\;\beta_{µ,\;0}+\beta_{µ,\;1}X+\sum_{i=2}^{q}\beta_{µ,\;i}Q_{i}\left(x\right)$ $\sigma^{*}\left(x\right)=\beta_{\sigma^{*},\;0}+\beta_{\sigma^{*},\;1}X+\beta_{\sigma^{*},\;2}Y+\sum_{i=3}^{q}\beta_{µ,\;i}Q_{i}\left(x\right)$	1
Schlather **(M2)**	$µ\left(x\right)=\;\beta_{µ,\;0}+\beta_{µ,\;1}X+\sum_{i=2}^{q}\beta_{µ,\;i}Q_{i}\left(x\right)$ $\sigma^{*}\left(x\right)=\beta_{\sigma^{*},\;0}+\beta_{\sigma^{*},\;1}X+\beta_{\sigma^{*},\;2}Y+\beta_{\sigma^{*},\;3}X*Y+\sum_{i=4}^{q}\beta_{µ,\;i}Q_{i}\left(x\right)$	1.007
Schlather **(M3)**	$µ\left(x\right)=\;\beta_{µ,\;0}+\beta_{µ,\;1}X+\sum_{i=2}^{q}\beta_{µ,\;i}Q_{i}\left(x\right)$ $\sigma^{*}\left(x\right)=\beta_{\sigma^{*},\;0}+\beta_{\sigma^{*},\;1}X+\beta_{\sigma^{*},\;2}Y+\sum_{i=3}^{q}\beta_{µ,\;i}Q_{i}\left(x\right)$	1.022
Brown–Resnick **(M4)**	$µ\left(x\right)=\;\beta_{µ,\;0}+\beta_{µ,\;1}X+\sum_{i=2}^{q}\beta_{µ,\;i}Q_{i}\left(x\right)$ $\sigma^{*}\left(x\right)=\beta_{\sigma^{*},\;0}+\beta_{\sigma^{*},\;1}X+\beta_{\sigma^{*},\;2}Y+\beta_{\sigma^{*},\;3}X*Y+\sum_{i=4}^{q}\beta_{µ,\;i}Q_{i}\left(x\right)$	1.427

^1^ Estimated max-stable models under the pairwise likelihood approach; ^2^ trend surfaces denote the pointwise distribution imposed on each spatial unit where dengue case counts were collected. *X, Y* denote the horizontal and vertical coordinates, respectively, and *Q* refers to the spatial covariates as described in Section 3.1. No restrictions were imposed on the scale parameter *ξ*; ^3^ the relative deviance is given by the ratio of the pairwise deviance for the model of interest over the best model (M1).

**Table 3 viruses-14-02450-t003:** Characteristics of the 1618 block maxima by year.

	Year (20XX ^1^)													
	07	08	09	10	11	12	13	14	15	16	17	18	19	20
Min.	0	0	0	0	0	0	0	0	0	0	0	0	0	0
25th perc.	0	0	0	0	0	0	1	0	0	0	0	0	0	1
50th perc.	1	1	1	1	1	1	1	1	1	1	0.5	0	1	2
75th perc.	1	1	1	1	1	1	3	2	2	2	1	1	2	4
Max.	27	16	13	24	19	20	31	107	26	26	31	21	39	42

^1^ XX refers to the last two numerals indicating the year (e.g., 07, 08, …, 20).

**Table 4 viruses-14-02450-t004:** Independent effects of per-unit change on the 30-year return levels under M1.

Spatial Factor	Change in Return Level ^1^	Average Return Level % Change
3-year increase in Median age of public apartments	1.8	3.8%
10% increase in Impervious surfaces	1.6	3.3%
Vegetation ^2^	0.7	1.4%
Freshwater surfaces	0.3	0.7%
Population size	0.1	0.3%

^1^ Denotes the respective change in 30-year return levels on average over each spatial unit; ^2^ with human management and tree canopy.

## Data Availability

All computational codes used in this study are available from https://github.com/stacysoh/SEVT (accessed on 29 August 2022). The data are not publicly available due to the data restriction policy.

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
