# Peer review of "Spatial Methods for Inferring Extremes in Dengue Outbreak Risk in Singapore"

_viruses, 2022, doi:10.3390/v14112450_

Round 1
Reviewer 1 Report
Dear Authors,
Please find the suggestion in attachment!
Best,

Reviewer 2 Report
Spatial Methods to Infer Extremes in Dengue Outbreak Risk in Singapore
In the manuscript, the authors examined the spatial distribution of extreme weekly dengue outbreak risk in Singapore from 2007 to 2020 and divided Singapore into equal-sized hexagons. They used a specific and non-usual analysis method and among their main findings was that the presence of a high concentration of public old apartments as well as impervious surfaces represented the large impacts of the outbreak of dengue in Singapore.
I only have minor comments
Line 29-35: insert some references into the text.
Lines 50-55_In the context of infectious diseases and modeling their extremes across space can help policymakers formulate targeted interventions in areas with exceptionally high risk, especially when resources for interventions are limited during an outbreak, which is important for long-term resource planning and control efforts.
- Some phrases are too extensive and should be shortened.
Line 90-91 - Epidemiological information such as the residential ad-90 dresses and onset dates of all laboratory-confirmed dengue cases were anonymized.
Please insert the ethical committee number and data into the manuscript.
Line 110: We assumed that the land cover types remained constant throughout the study period.
- At a 13-year interval, how may the authors assume that all the land surface types analyzed remained the same?
298-215 Relocate for Material and Methods.
426-428_ EVT is gaining currency in infectious disease modeling. It has been applied to 426 weekly rates of pneumonia and influenza (23), the historical distribution of pandemics (6), as well as super spreading events for respiratory events (33).
The manuscript should be submitted to the English edition such as superspreading events for respiratory events.
Reviewer 3 Report
This paper proposes a spatial model to infer extremes in dengue outbreak in Singapore. The overall structure has been designed and presented very well. There are a couple of minor comments.
Instead of one sentence to summarize, it might be good to review some other relevant research on extreme value theory and infectious disease study in order to highlight the importance of this study. It also applies to spatial models and disease study.
Map design could be improved. It is not easy to tell the areas of interests with all the same small maps.
Supplementary materials are not available.
